# Broad-Spectrum Bactericidal Activity of a Synthetic Random Copolymer Based on 2-Methoxy-6-(4-Vinylbenzyloxy)-Benzylammonium Hydrochloride

**DOI:** 10.3390/ijms22095021

**Published:** 2021-05-09

**Authors:** Anna Maria Schito, Gabriela Piatti, Debora Caviglia, Guendalina Zuccari, Silvana Alfei

**Affiliations:** 1Department of Surgical Sciences and Integrated Diagnostics (DISC), University of Genoa, Viale Benedetto XV, 6, I-16132 Genova, Italy; amschito@unige.it (A.M.S.); gabriella.piatti@unige.it (G.P.); Caviglia86@gmail.com (D.C.); 2Department of Pharmacy, University of Genoa, Viale Cembrano, 16148 Genoa, Italy; zuccari@difar.unige.it

**Keywords:** multidrug-resistant bacteria, cationic antibacterial copolymer, 2-methoxy-6-(4-vinyl benzyloxy)-benzylammonium hydrochloride, membrane permeabilization, Gram-positive and Gram-negative bacteria, MIC and MBC determination, time–kill experiments, turbidimetric studies

## Abstract

Low-molecular-weight organic ammonium salts exert excellent antimicrobial effects by interacting lethally with bacterial membranes. Unfortunately, short-term functionality and high toxicity limit their clinical application. On the contrary, the equivalent macromolecular ammonium salts, derived from the polymerization of monomeric ammonium salts, have demonstrated improved antibacterial potency, a lower tendency to develop resistance, higher stability, long-term activity, and reduced toxicity. A water-soluble non-quaternary copolymeric ammonium salt (P7) was herein synthetized by copolymerizing 2-methoxy-6-(4-vinylbenzyloxy)-benzylammonium hydrochloride monomer with *N, N*-di-methyl-acrylamide. The antibacterial activity of P7 was assessed against several multidrug-resistant (MDR) clinical isolates of both Gram-positive and Gram-negative species. Except for colistin-resistant *Pseudomonas aeruginosa*, most isolates were susceptible to P7, also including some Gram-negative bacteria with a modified charge in the external membrane. P7 showed remarkable antibacterial activity against isolates of *Enterococcus*, *Staphylococcus, Acinetobacter*, and *Pseudomonas*, and on different strains of *Escherichia coli* and *Stenotrophomonas maltophylia*, regardless of their antibiotic resistance. The lowest minimal inhibitory concentrations (MICs) observed were 0.6–1.2 µM and the minimal bactericidal concentrations (MBC) were frequently overlapping with the MICs. In 24-h time–kill and turbidimetric studies, P7 displayed a rapid non-lytic bactericidal activity. P7 could therefore represent a novel and potent tool capable of counteracting infections sustained by several bacteria that are resistant to the presently available antibiotics.

## 1. Introduction

Small-molecular-weight quaternary ammonium salts (QASs), such as the well-known benzalkonium hydrochloride, commercially available as CITROSIL^®^ [1], and several of its derivatives, have been broadly studied and intensively used as antimicrobial agents, endowed with considerable broad-spectrum potency against Gram-positive and Gram-negative bacteria and some fungi [2]. In general, quaternary ammonium derivatives act as ionic detergents, capable of rapidly binding to the negatively charged bacterial wall and membranes [2]. After absorption on the bacterial surface, intercalating with the membranes and altering their normal structure, QASs cause the formation of pores, modifying membranes’ permeability, thus causing the loss of enzymes, coenzymes, and ions. Due to these events, QASs compromise the biosynthetic activities of bacteria, triggering their death [2]. Unfortunately, QASs’ non-specific antimicrobial mechanism translates into low selectivity for pathogens, significant toxicity toward eukaryotic cells, and hemolytic toxicity [2,3,4]. These drawbacks significantly hamper their administration in vivo by the oral route and/or intravenous injection, limiting their use as surface disinfectants and in epidermal treatment [2]. However, although the side effects caused by the topical application of QASs are relatively moderate in comparison to those derived from systemic administrations [5], a number of limitations concerning the topical use of low-molecular-weight QASs have also become noticeable.

The strongly cationic and highly water-soluble structure of QASs is subjected to uncontrollable diffusion, sustained by QASs’ low molecular weight, which results in short-term functionality on target surfaces [6] and the need to increase the dosage [7].

On the other hand, when less cationic small antimicrobial agents with amphiphilic structures ere topically administered, undesired permeation and penetration can occur, causing transdermal delivery. Consequently, unnecessarily burdening of the systemic circulation [5] and damage to organs, causing local allergic contact dermatitis (ACD) on the skin [8] or drug-induced liver injury (DILI) [9], can develop.

A recognized and effective strategy used to overcome these issues consists in providing the small QASs with polymerizing groups, obtaining active cationic monomers and in transforming them, through polymerization reactions, into macromolecular structures, in which the cationic groups are incorporated into the polymer backbones [10]. The polymeric quaternary ammonium salts (PQASs) thus obtained have been demonstrated to possess improved antibacterial potency, less of a tendency to develop resistance, higher stability, long-term activity, and reduced toxicity [11]. Collectively, the antimicrobial effects can be maintained and even improved due to the increased local density of cationic groups allowed by the multivalence of the polymer [11]. Simultaneously, the polymer architecture attenuates the excessive cationic and polar character of the small molecules, which could also be detrimental for mammalian cell membranes, thus reducing the hemolytic toxicity and cytotoxicity [11,12].

The possible excessive residual cationic character present in homo-polymers, causing outstanding hemolytic toxicity, could be further toned down by diluting the active QAS monomers with uncharged comonomers. Through this approach, random or block copolymers with tunable selectivity and modular antimicrobial effects based on the ratio of cationic to uncharged residues can be obtained [11,12,13]. In this regard, several types of monomers containing permanently cationic tetra alkyl ammonium groups have been developed and widely employed to prepare homopolymers and copolymers that are able to inhibit bacteria growth and/or to kill them simply on contact [11,14,15].

A new trend in the design of antimicrobial polymers involves employing monomers containing primary ammonium groups in the form of acidic salts to mimic the amphiphilic properties and cationic functionalities of antimicrobial poly-lysine [11,14,15,16].

Comparisons between macromolecules containing primary ammonium groups and those possessing quaternary ones revealed that the polymers bearing primary ammonium groups outperformed their tertiary and quaternary analogues, in terms of having more potent antibacterial activity and lower toxicity [11,13,16]. Furthermore, the preparation of random copolymers, merging such monomers with uncharged acrylates, methacrylates, acrylamides and methacrylamides, allowed researchers to modulate the balance between hydrophobic and cationic properties (HLB), and to control the polymer length [16].

Fast inhibitory effects, high selectivity for pathogens, and a limited tendency to develop resistance are significant advantages associated with the use of cationic polymers and copolymers [11,13,17].

Generally, the selectivity of cationic macromolecules towards bacterial membranes depends on fundamental differences in the cell membrane lipid composition and surface components [13,17,18,19,20]. Since similar differences have also been observed in the membrane of cancer cells, cationic peptides [21], polymers, and copolymers [20] have been reported to exert significant cytotoxic activity against several type of tumors, regardless of the resistance of cells to conventional therapies such as etoposide (ETO). Moreover, it is generally accepted that the very low propensity to induce resistance displayed by these cationic macromolecules is derived from their non-specific mechanism of action. In virtue of this, the cationic agents do not necessarily need to enter the cell and target enzymatic processes, which can mutate and develop resistance, but rather, they are capable of killing the target rapidly, simply on contact [11,13,22,23,24]. Consequently, widespread and rapid resistance development towards membrane-active and cell lysing antimicrobial macromolecules is unlikely compared to conventional antimicrobials.

Specifically, random copolymers have positively charged regions and neutral fractions randomly isolated along the polymer backbone.

In this context, we recently reported the preparation, physicochemical characterization, and antibacterial properties of a cationic water-soluble copolymer (P5) (Figure 1), obtained by polymerizing the 4-ammoniumbuthylstyrene hydrochloride monomer (M5) with di-methyl acrylamide (DMAA) used as a comonomer [13].

P5 proved to be much more potent than the low-molecular-weight monomer M5 and displayed remarkable antibacterial activity against most of the 61 isolates tested, belonging both to Gram-positive and Gram-negative species [13]. Since it is known that the cytoplasmic membrane of cancer cells, including neuroblastoma (NB) cell lines, is more like that of prokaryotes than that of eukaryotes, and cationic material active on bacteria are selectively cytotoxic also on tumor cells, including multidrug-resistant ones [20,21,25], we recently observed the cytotoxic effects of P5 on two cell lines that were differently sensitive to ETO [20]. In that study, we also reported the preparation, physicochemical characterization, and the significant cytotoxic effects of a new water-soluble cationic copolymer (P7) (Figure 2), obtained by polymerizing 2-methoxy-6-(4-vinylbenzyloxy)-benzylammonium hydrochloride monomer (M7) with di-methyl acrylamide (DMAA) comonomer [20].

Based on the results obtained with P7, which showed strong reactive oxygen species (ROS)-related cytotoxic effects on NB cells and particularly on those resistant to ETO (HTLA-ER cells), in the present study we aimed to evaluate its cytotoxic effects on bacterial cells. In this work, the antibacterial activity of P7 was first assessed against many Gram-positive and Gram-negative MDR nosocomial isolates by determining MIC and MBC values. Subsequently, P7′s biocidal properties and the mechanism of action by which P7 kills bacterial cells were investigated on opportunely selected strains of both species by performing time–kill and turbidimetric experiments.

## 2. Results and Discussion

### 2.1. Synthesis and Spectrophotometric Characterization of 2-Methoxy-6-[(4-Vinyl)Benzyloxy]Benzylammonium Hydrochloride M7 (7)

M7 was prepared by performing the synthetic procedure reported and extensively discussed in a recent study [20]. The experimental details are also included in Appendix A associated with this article. The synthetic intermediates and the final compound M7 were spectrophotometrically characterized by means of FTIR and NMR analyses, as shown in the Appendix A, which confirmed their structures, according to what has been reported in the literature [20]. In addition, other analysis, including HPLC, GC-MS, elemental analysis, as well as melting and boiling point determinations, were carried out when useful or necessary and the results, which are also available in Appendix A, were successfully compared with those that have previously been reported [20].

### 2.2. Preparation of Copolymer P7 by Radical Copolymerizations in Solution and Spectroscopic characterizations of P7

Copolymer P7 was obtained according to a recently reported and discussed procedure [20], which was also included in Appendix A). Particularly, M7 was copolymerized with DMAA in the presence of azobisisobutyronitrile (AIBN) as a radical initiator, in MeOH at 60 °C, for 72 h, achieving a conversion of 85%. P7 was characterized by several techniques, as reported in Appendix A). However, for the sake of clarity, the main physicochemical properties of P7 have been included in Table 1 below.

A detailed discussion of the results reported in Table 1 is available in Alfei et al. (2021) [20].

### 2.3. Antibacterial Properties

#### 2.3.1. The Reasons for Our Interest in Monomer M7 and Copolymer P7

Having shown remarkable cytotoxicity on tumor cells with membranes resembling those of bacteria in terms of charge [20], and differing from those of eukaryotic cells, towards which cationic materials such as P7 have been reported to be non-cytotoxic [21,25], the P7 copolymer appeared to us as an excellent candidate for developing a new antibacterial tool for several reasons. Due to the structure of M7, used as a cationic monomer, P7 possessed primary benzylammonium hydrochloride residues as cationic moieties, which have been reported to confer interesting antimicrobial properties on several homopolymers and copolymers [11,26,27,28,29]. In addition, the primary amino group, in the form of hydrochloride salt with cationic functionality, was considered preferable to the permanently protonated quaternary ammonium groups because, among the various studies published, the macromolecules containing the quaternary benzylammonium group were less effective than those not containing quaternary ammonium residues, providing reversible cationic groups [12]. Accordingly, we recently reported the broad-spectrum bactericidal effects of a polystyrene-based copolymer containing a primary ammonium group as a cationic function, characterized by a C4 carbon chain as a spacer between the cationic groups and the phenyl rings [13]. Moreover, several copolymers were synthetized containing primary amine groups in the form of acid salts, which proved to possess considerable broad-spectrum antimicrobial properties, often superior to those of the quaternary ammonium analogues [11]. P7 attracted our attention as a probable antibacterial cationic copolymer because, although antibacterial polymers obtained by polymerizing or copolymerizing 4-aminomethylstyrene monomers are well documented, to our knowledge no studies have investigated the use of benzylamine monomers containing the polymerizable styrene moiety as a substituent on the aromatic ring containing the active benzylamine cationic group. In addition, we noted that P7, due to the structure of M7, that contain di-alkyl ether groups in *meta*-positions respectively, possessed residues like those present in the aromatic ring of vancomycin. Indeed, vancomycin owns di-phenyl ether groups in *meta*-position each other, and is indicated for the treatment of serious, life-threatening infections by Gram-positive bacteria unresponsive to other antibiotics (MIC < 4 µg/mL) [30]. Moreover, P7 possessed 1,2,3-tri-substituted phenyl residues, having ether groups in positions 2 and 6 and containing the CH_3_O ether group. In this regard, P7 reproduces, in its structure, that of natural substances made of aromatic rings 1,2,3-tri-substituted with two CH_3_O groups in positions 2 and 6, such as combretastatin E, which has displayed MIC values lower or comparable to those of ampicillin and chloramphenicol against *S. aureus, P. aeruginosa*, and *E. faecalis* [31]. Note that ampicillin is classified by the World Health Organization (WHO) as critically important for human medicine [32], whereas chloramphenicol is on the WHO’s List of essential medicines [33]. Finally, in view of a possible clinical application of P7, its high water-solubility would assure easy routes of administration and high bioavailability.

#### 2.3.2. Antimicrobial Activity of P7 by Determination of MIC and MBC Values

MIC and MBC values for P7 were obtained by analyzing a total of 61 strains of clinical origin, including MDR isolates of both Gram-positive and Gram-negative species and Gram-negative strains with modifications in the outer membrane, therefore having a reduced total negative surface charge.

For comparison, monomer M7 (**7**) was also analyzed under the same conditions. Although against various Gram-positive and Gram-negative strains M7 showed significantly lower MICs than those of the recently published cationic styrene monomer [13], accordingly to several articles [11,13,27,28,29,34,35] M7 was considered ineffective against all isolates evaluated in this study. On the contrary, the macromolecular compound P7 demonstrated very interesting results both on Gram-positive (Table 2) and Gram-negative species (Table 3), inhibiting the growth of bacteria regardless of their multidrug resistance pattern. In this regard, it has generally been accepted that cationic macromolecules, possessing a non-specific mechanism of action, do not need necessarily to enter the cell and target enzymatic processes to which bacteria have mutated, developing resistance. Therefore, they are also capable of killing MDR isolates rapidly, simply on contact, thus overcoming their acquired ability to resist traditional antibiotics [11,13,22,23,24].

Interestingly, the MIC values of P7 against Gram-positive isolates were very similar to those obtained by the recently published P5 copolymer [13]. However, differently from what was observed when testing its anti-tumor activity on NB cells, which was lower than that of P5 [20], when tested on bacteria P7 was found to be more cytotoxic than P5 [13], showing both MIC and MBC values very much lower than those of P5. The lowest MIC values were observed against the genus *Enterococcus*, including some vancomycin-resistant strains (VRE) (MICs = 0.6–1.15 µM on *E. faecium* and 2.3 µM on *E. faecalis*) and against the sporogenic *Bacillus subtilis* (MIC = 1.15 µM), which reported double or overlapping MBC values compared to the MICs. Concerning the *Staphylococcus* genus, in which methicillin-resistant strains (MRS) were also included, the range of MIC values was slightly wider (MICs = 0.6–4.6 µM). P7 proved to be far more potent than three random cationic copolymers (namely PAI1–PAI3) containing alkyl ammonium hydrochloride moieties and aromatic rings like P7, against methicillin-resistant *S. aureus* (MRSA) (MIC = 14.9 µM (PAI2), 17.7 µM (PAI3) and 267.8 µM (PAI1)) [36]. Against *S. epidermidis*, P7 was slightly less effective than PAI2, equally active to PAI3 and more potent than PAI1. With regard to bacterial strains with modified membrane charges, which are usually non-susceptible to antibacterial cationic peptides and cationic macromolecules [37,38,39,40,41] and against which the previously reported copolymer P5 was totally ineffective, P7 displayed significant activity, particularly against *Yersinia enterocolitica* (strain 342) and *Providencia stuartii* (strain 374) (MIC = 9.3 µM).

Curiously, P7 was slightly less active than P5 against *Klebsiella pneumoniae* 259 but showed lower MIC values against all *Klebsiella* (4.6–9.3 µM) and *Pseudomonas* (2.3–9.3 µM), and against all the strains of *Escherichia coli* (2.3–4.6 µM), all isolates of *Acinetobacter baumannii* (2.3–9.3 µM), and *Stenotrophomonas maltophylia* (2.3–9.3 µM), as well as against the *Salmonella* gr. B strain (4.6 µM).

The MICs observed for P7 on *E. coli* (2.3–4.6 µM) were in a narrower and lower range than those of the self-degradable antimicrobial copolymers (P4, P6, P7, and P9) prepared by Mizutani and colleagues (2012) [42], bearing similar cationic groups in the form of primary ammonium salts.

In a recent study by Wen and colleagues, it was reported the synthesis, characterization, and antibacterial properties of four cationic nanosized copolymers (CNPS 1–4), prepared by copolymerizing different amounts of the quaternary cationic monomer N-[2-(methacryloyloxy)ethyl]-N,N-dimethyltetradecane-1-ammonium bromide (MDTP) and styrene as a comonomer [26]. The study showed that the antibacterial activity of CNPS increased with the increase in the content of MDTB in the formulae. Furthermore, regarding the best-performing copolymer, CNPS-4, containing 80% MDTP and with ζ-p of +58 mV, the MIC values reported on *E. coli* and *S. aureus* (both ATCC reference strains) were 48.8 and 25.0 µM, respectively.

These results confirmed that the increase in cationic groups in macromolecular formulas, and consequently the increase in the density of positive charges on the surface of polymers, improves the interaction between the cationic (co)polymers and the negatively charged bacteria, leading to serious consequences for bacteria, such as growth inhibition and death [26].

Based on our results, P7 was shown to be 5–10 times more active than CNPS-4 against clinical *E. coli* isolates and more than 10 times active against *S. aureus* (MIC (P7) = 4.6 µM vs 50 µM of CNPS-4), although P7 contained a smaller fraction of cationic monomer (32 mol%) and had a lower ζ-p and charge density than CNPS-4. To obtain greater antibacterial activity, the winning strategy of adopting primary amino groups in the form of hydrochloride salts, instead of the widely used quaternary ammonium group, therefore appeared to be confirmed.

In addition, when compared to another of the copolymers published by Wen (i.e., CNPS-3), which possesses the same surface electrical charge as that of P7 (ζ-p = + 50 mV) P7 was found to be 44–88 times more active against *E. coli* and 35 times more active on *S. aureus.*

Regarding Gram-positive species, P7 was much more active against MRSA isolates (4.6 µM *vs* > 7.4 µM) and up to 12 times more active against *E. faecium* than the ACP1Gly molecule, which belongs to a family of cationic amino-acid-modified polymers recently prepared by Barman et al. (2019) [43].

As for its antibacterial effects against *B. subtilis, S. aureus*, and *E. faecium*, P7 proved to be more potent than the homopolymer (Poly**1**), containing benzylamine residues, reported by Gelman et al. (2004) [12]. P7 displayed MIC values (µM) that were 4.3 times lower against *B. subtilis*, 4.3–8.3 times lower against *E. faecium*, and values that were two times lower against *S. aureus*. Against *E. coli*, P7 displayed MIC values (µM) that were 2–4 times lower than those of Poly**1**. Like the cationic antibacterial copolymer, we recently reported [13], P7 was ineffective against a colistin-resistant strain of *P. aeruginosa*, due to a reduced possibility of electrostatic interactions with the outer membrane of the isolate, caused by an adaptive modification of lipid A [13,37]. Additionally, P7 was found to be more active than a derivative of the potent natural cationic peptide magainin II, known as Ala^8,13,18^-magainin 2 amide, and which has been reported to exhibit potent antimicrobial activity.

Regarding the Enterobacteriaceae family, although P7 was less active than the cationic polymer ACP1Gly [43] against *E. coli*, it was more active against some isolates of KPC-producing *K. pneumonia*. Furthermore, Weiyang et al. (2018) [44] reported the remarkable antibacterial activity of two types of polyionenes, namely 2a (Mn = 5362) and 2b (Mn = 5281), against 20 clinical strains of *K. pneumoniae* which are responsible for lung infections, and also associated with faster killing kinetics than imipenem and other commonly used antibiotics. Although the MICs displayed by the two polymers were in the range of 1.5–6.0 µM(2a) and in the range of 6.0–48.5 µM (2b) on *K. pneumoniae*, P7, which displayed MICs values in the range of 4.6–9.3, proved to be less active than 2a, but far more active than 2b, with an activity that was independent of the diversity between the strains. In the same study, the MIC values of 2a and 2b were also determined against one strain of *E. coli* (MICs = 3.0 µM (2a) and 24.2 µM (2b)), one MRSA isolate (1.5 µM (2a) and 24.2 µM (2b)) and on representative strains of non-fermenting Gram-negative bacterial species, such as *A. baumannii* (6.0 µM (2a) and 24.2 µM (2b)) and *P. aeruginosa* (6.0 µM (2a) and 48.4 µM (2b)). Based on these results, P7 was slightly less active than 2a on *E. coli* and significantly less active on MRSA, but more active against both *A. baumannii* and *P. aeruginosa*, also considering the greater number of isolates tested by us. If compared to 2b, P7 was much more powerful against all these species. The scope of this study was to develop a new antibacterial molecule that was more effective than available antibiotics against MDR bacteria. Consequently, in Table 4, a comparison is reported between the MIC values observed for P7 on the MDR isolates of the different species tested here and the MIC values obtained by the antibiograms of the same strains against commonly used antibiotics.

The data reported in Table 4 demonstrate that P7 can be considered an efficient antibacterial agent, capable of inhibiting numerous MDR pathogens, towards which many of the current antibiotics are no longer active. Furthermore, it can be observed that, although the MIC values of P7 against different MDR isolates of the same species have very homogeneous values or fall within a narrow range of concentrations, those relating to antibiotics are less homogeneous and denote strain-dependent variability.

#### 2.3.3. Time–Kill Curves

Time–kill experiments were carried out using P7 at concentrations four times the MIC values on three isolates of *P. aeruginosa*, two of *K. pneumoniae*, and three of *S. aureus*. Figure 3 shows the most representative curves obtained for each species, which established that P7 possessed a very strong bactericidal effect on all the assayed pathogens. Indeed, a rapid decrease of 3.8–4.1 logs in the original cell number was evident already after 30 min of exposure to P7, with *P. aeruginosa* being the most susceptible species. After two hours of exposure, a total decrease in the original cell number occurred, regardless of the bacterial species tested. During the next two hours, a slightly and not significant regrowth for all the species was observed, particularly for *K. pneumoniae*. In the subsequent period up to 24 h, the number of *K. pneumoniae* colonies remained practically unchanged, and no further regrowth was observed; those of *S. aureus* cleared after 6 h and did not undergo further regrowth, whereas the colonies of *P. aeruginosa* underwent a progressive decrease until complete elimination after 24 h.

Overall, no significant regrowth was observed after 24 h of incubation with P7 for all strains of the three species tested. Interestingly, this mechanism of action overlapped that of the P5 copolymer that we have recently reported [13], even though it is structurally different (P7 was in fact prepared from a monomer (M5) having styrene group units as polymerizing units and through copolymerization with DMAA). To the best of our knowledge, cases in which the bactericidal behaviors of cationic materials last up to 24 h are rarely reported in the literature. Commonly, for cationic antibacterial molecules, time–kill experiments are often not reported for more than 2, 4, 6, or 8 h [43,44,45]. In fact, for cationic bactericidal peptides, such as colistin [46] and dendrimers [24], for which data have been reported at 24 h, although rapid killing occurred even after just 5 min [46] and 1 h [24] after contact with the antibacterial device, an abundant bacterial regrowth arose after 24 h of action. We conclude that, since both P5 and P7 were obtained by polymerizing structurally very different cationic monomers using the same fraction of the same comonomer (DMAA), the above behavior, which is rarely observed for cationic macromolecules, could be related to the presence of DMAA moieties in the P5 and P7 backbone.

#### 2.3.4. Effect of P5 on the Growth Curve of *P. Aeruginosa*, *K. Pneumoniae*, and *S. Aureus*

The kinetics of growth in MH broth of the selected strains of *P. aeruginosa, K. pneumoniae*, and *S. aureus* used in time–kill studies were monitored at 600 nm for 6 h in the absence and presence of P7, which was used at a concentration of 4 × MIC. Figure 4 illustrates the results obtained for one representative strain of *P. aeruginosa, K. pneumoniae*, and *S. aureus.*

Although, as expected, the control cultures showed an exponential turbidimetric increase, the presence of P7 resulted in complete inhibition of growth. In a recent study on the antibacterial cationic copolymer P5 [13], the turbidimetric experiments did not show any decrease in optical density for all the isolates examined, thus suggesting a non-lytic bactericidal mechanism, regardless of the different species. On the contrary, in this study, the exposure of Gram-positive *S. aureus* (strains 18, 189, and 195) to P7 (Figure 4b) highlighted a different trend and a progressive reduction in the OD 600 values during the 6 h of the experiment. Interestingly, the decrease in optical density for the tested isolates showed a strain-dependent intensity.

Since bacterial lysis should have led to an immediate and much faster reduction in absorption at 600 nm, hypothesizing a lytic-type mechanism would be unfounded. Therefore, we speculated that these phenomena could be due to a tendency of the Gram-positive bacteria to form aggregates.

In this regard, Mukherjee and colleagues assessed the effects of exposing *E. coli* and *B. subtilis* to cationic antibacterial polymers based on side-chain amino acids. *E. coli*, when exposed to the cationic homopolymer based on leucine, has undergone a notable change in the morphology of its cells, which have changed during the treatment from a rod shape to a spherical shape. However, *B. subtilis* cells did not show any evident morphological changes, although in both cases an aggregation of cells was observed [47].

Collectively, the experiments by Mukherjee confirm that the optical density decrement observed by us during the treatment of Gram-positive *S. aureus* with P7 could be due to the tendency of this species to form aggregates.

Furthermore, it is rational to advance the hypothesis that the cationic P7 could behave like other CAPs and the recently reported P5 copolymer, i.e., causing damage to the bacterial membrane due to its electrostatic interaction with the negatively charged bacterial surface [11,13,22,48]. The inactivity of P7 on the colistin-resistant strain of *P. aeruginosa* included in our study and its reduced activity against isolates with reduced superficial negative charge, such as *Y. enterocolitica* (strain 342)*, P. stuartii* (strain 374)*, M. morganii* (strain 372), *S. marcescens* (strain 228), and *P. mirabilis* (strain 254) would confirm these hypotheses.

Regarding the non-lytic bactericidal activity of P7 on Gram-negative species, as with that of P5 and other previously reported cationic dendrimers [13,24], this fact could depend on the absence of the *N*-terminated hydrophobic fatty acid side chain, which is present in colistin and reported to be the main cause of its lytic behavior. However, the absence of a similar fatty acid side chain in P7, as in P5 [13], which enhances their hydrophilic character, could limit their diffusion towards the cytoplasmic membrane (CM) and consequently prevent cell lysis. Therefore, despite being bactericidal (inducing irreparable and lethal alterations of the membrane), they lack lytic properties.

## 3. Materials and Methods

### 3.1. Chemicals and Instruments

Monomer M7 and copolymer P7 were prepared according to recently reported procedures [20], which have been described in detail in the Appendix A, respectively, including Appendix A. The physicochemical characterization of M7 and P7 was carried out by performing the same analysis previously described and using the same instruments reported in recent studies [13,20] and the details of this process are available in the Appendix A.

### 3.2. Microbiology

#### 3.2.1. Microorganisms

A total of 61 isolates, belonging to several Gram-positive and Gram-negative species, were used in this study. All were clinical strains from a collection isolated by the clinical laboratories of the University of Genoa (Italy), and identified by VITEK^®^ 2 (Biomerieux, Firenze, Italy) or the matrix-assisted laser desorption/ionization time-of-flight (MALDI-TOF) mass spectrometric technique (Biomerieux, Firenze, Italy). Of the 23 Gram-positive organisms tested, ten strains belonged to the *Enterococcus* genus, (four *Enterococcus faecalis* (resistant to vancomycin (VRE)), three *E. faecium* VRE, one *E. casseliflavus* (intrinsically resistant to vancomycin), and one *E.durans* and one *E. gallinarum* (intrinsecally resistant to vancomycin)); 12 strains pertained to the *Staphylococcus* genus, including two methicillin-resistant *S. auresus* (MRSA) and one susceptible, three methicillin-resistant *S. epidermidis* (MRSE), two of which were also resistant to linezolid, one methicillin-resistant (MR) *S. haemolyticus*, one *S. hominis* MR, one *S. lugdunensis*, one *S. sapropyticus*, one *S. simulans* MR, and one *S. warneri.* A strain of the sporogenic *Bacillus subtilis* was also added. Regarding the thirty-eight Gram-negative isolates, 18 strains were *Enterobacteriaceae*: three *Escherichia coli* (one was susceptibile to all antibiotics tested and one was a O157:H7 strain), one *Proteus mirabilis*, one *Morganella morganii*, one *Providencia stuartii*, one group B *Salmonella*, one *Serratia marcescens*, one *Yersinia enterocolitica*, six group A carbapenemase-producing *Klebsiella pneumoniae*, two non-carbapenemase-producing *K. pneumoniae*, and one *K. oxytoca.* Twenty strains belonged to the non-fermenting group: six *Pseudomonas aeruginosa*, one *P. fluorescens*, one *P. putida*, six *Stenotrophomonas maltophylia*, five *Acinetobacter baumannii*, and one *A. pittii.*

#### 3.2.2. Determination of MIC and MBC

To investigate the antimicrobial activity of M7 and P7 on the 61 pathogens, their minimal inhibitory concentrations (MICs) were determined by following the microdilution procedures detailed by the European Committee on Antimicrobial Susceptibility Testing (EUCAST) [49].

Briefly, after overnight incubation, cultures of bacteria were diluted to yield a standardized inoculum of 1.5 × 10^8^ CFU/mL. Appropriate aliquots of each suspension were added to 96-well microplates containing the same volumes of serial 2-fold dilutions (ranging from 1 to 512 μg/mL) of M5 or P5 to yield a final concentration of approximately 5 × 10^5^ cells/mL. The plates were then incubated at 37 °C. After 24 h of incubation at 37 °C, the lowest concentration of M7 or P7 that prevented a visible growth was recorded as the MIC. All MICs were obtained in triplicate, the degree of concordance in all the experiments was 3/3, and the standard deviation (±SD) was zero. The minimal bactericidal concentration (MBC) has been defined as the lowest concentration of a drug that results in killing 99.9% of the bacteria being tested [50].

The MBCs of M5 and P5 on the 61 pathogens were determined by subculturing the broths used for MIC determination. Ten microliters of the culture broths of the wells corresponding to the MIC and to higher MIC concentrations where plated onto fresh MH agar plates, and further incubated at 37 °C overnight.

The highest dilution that yielded no bacterial growth on the agar plates was taken as the MBC. All tests were performed in triplicate and the results were expressed as the mode.

#### 3.2.3. Time–Kill Curves

Time–kill curve assays for P7 were performed on three representative isolates of *P. aeruginosa* (strains 247, 256, and 259), two representative strains of *K. pneumoniae* (strains 366 and 369, both producing class A carbapenemases), and two representative isolates of *S. aureus* (strains 18 and 195, both MRSA), as previously reported [51]. All were clinical strains from a collection or isolated from volunteers. Experiments were performed over 24 h at P7 concentrations of four times the MIC for all strains.

A mid-logarithmic phase culture was diluted in Mueller–Hinton (MH) broth (Merck, Darmstadt, Germany) (10 mL) containing 4 x MIC of the selected compound in order to give a final inoculum of 1.0 × 10^5^ CFU/mL. The same inoculum was added to cation-supplemented Mueller–Hinton broth (CSMHB) (Merck, Darmstadt, Germany) as a growth control. Tubes were incubated at 37 °C with constant shaking for 24 h. Samples of 0.20 mL from each tube were removed at 0, 30 min, 2, 4, 6, and 24 h, diluted appropriately with a 0.9% sodium chloride solution to avoid carryover of P7 being tested, plated onto MH plates, and incubated for 24 h at 37 °C. Growth controls were run in parallel. The percentage of surviving bacterial cells was determined for each sampling time by comparing colony counts with those of standard dilutions of the growth control. Results have been expressed as log10 values of viable cell numbers (CFU/mL) of surviving bacterial cells over a 24 h period. A bactericidal effect was defined as a 3 log10 decrease of CFU/mL (99.9% killing) of the initial inoculum. All time–kill curve experiments were performed in triplicate.

#### 3.2.4. Evaluation of the Antimicrobial Effect of P7 through Turbidimetric Studies

The study of the antimicrobial activity of P7 was carried out by measuring the optical density variations (OD) as a function of time in cultures of the same strains employed for the time–kill experiments (three strains of *P. aeruginosa*, two of *K. pneumoniae*, and two of *S. aureus*) at a wavelength of 600 nm in a Thermospectronic spectrophotometer (Ultrospec 2100pro, Amersham Biosciences, Little Chalfont, UK) [52].

Bacterial cells were harvested from 10 mL of bacterial cultures in MH broth, and the cell number was adjusted to produce a heavy inoculum (OD adjusted to 0.2) corresponding to 10^8^ cells/mL. Cell suspensions were treated with or without P5 at concentrations equal to 4 MIC and incubated at 37 °C. After 30 min and 1, 2, 3, 4, 5, and 6 h of incubation, aliquots were taken from the cultures, and absorbance values were recorded at 600 nm. Measurements were blanked with MH broth containing an equivalent amount of P7 as that being tested. Experiments were performed in triplicate. The number of CFU was determined in parallel with the method described in the time–kill section and compared with the untreated sample.

## 4. Conclusions

In this study, it has been established that the hydrophilic cationic random copolymer P7, which has already been shown to possess potent ROS-related cytotoxicity against NB cells, and particularly against those resistant to etoposide, is also an excellent antibacterial compound, with bactericidal activity against various strains of different species belonging to both Gram-positive and Gram-negative bacteria. In time–kill experiments and turbidimetric studies, P7 very quickly killed MDR isolates that were representative of threatening pathogens, such as *S. aureus, K. pneumoniae*, and *P. aeruginosa*, with no regrowth observed even after 24 h of exposure. P7 was far more active than the recently reported copolymer P5 and susceptible bacteria were eradicated regardless of their antibiotic resistance at MIC values even lower than 1 µM. If, as reported in previous studies concerning similar cationic materials, P7 proves not to express toxicity for normal mammalian cells [21,25], and if it manifests favorable pharmacokinetic properties, thanks to its remarkable bactericidal action, to its favorable physicochemical characteristics, and its water solubility, this new copolymer could represent an innovative agent capable of successfully treating infections caused by various pathogens that are resistant to antibiotics currently available.

## Figures and Tables

**Figure 1 ijms-22-05021-f001:**
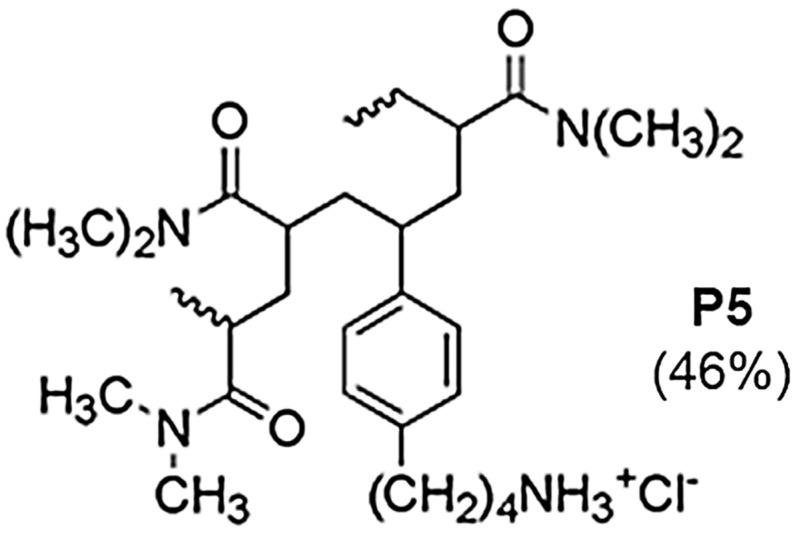
Structure of copolymer P5.

**Figure 2 ijms-22-05021-f002:**
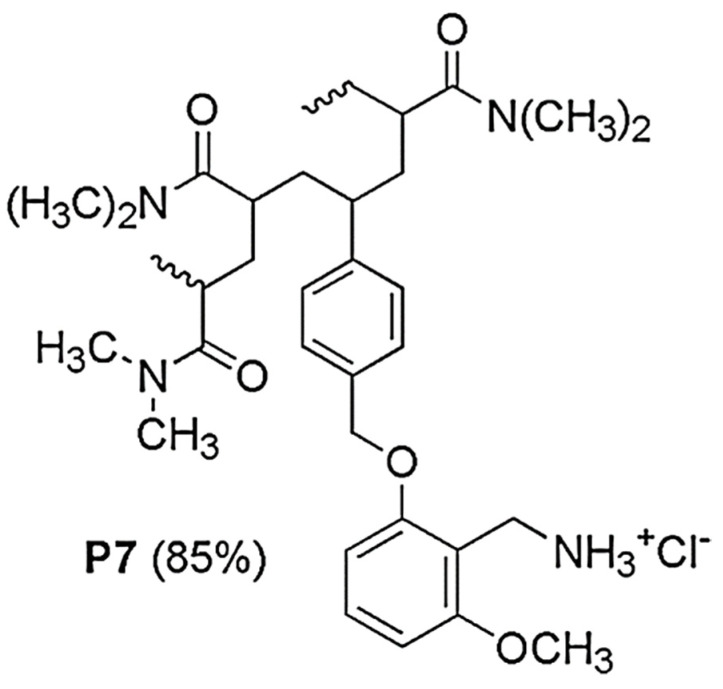
Structure of copolymer P7.

**Figure 3 ijms-22-05021-f003:**
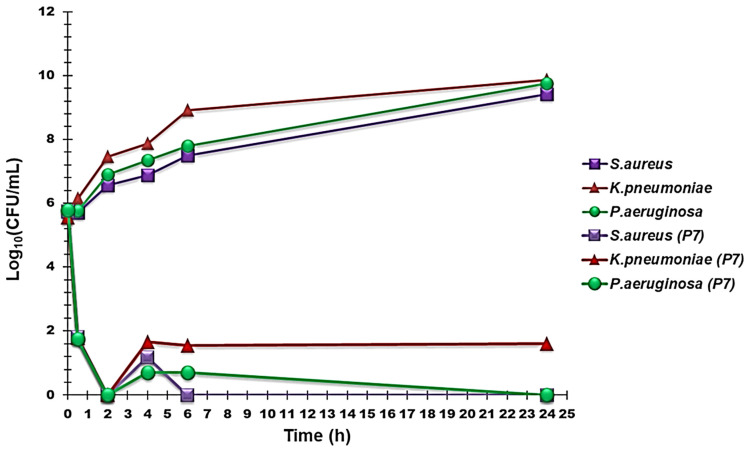
Time–kill curves performed with P7 (at concentrations equal to 4 × MIC) on *P. aeruginosa* 247, *K. pneumoniae* 366, and *S. aureus* 18.

**Figure 4 ijms-22-05021-f004:**
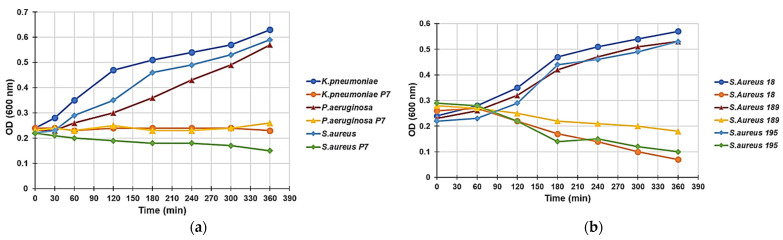
Effect of P7 (at 4 × MIC) on the growth curve of *K. pneumoniae* (strain 466), *S. aureus* (strain 18), and *P. aeruginosa* followed at 600 nm for a period of 6 h (**a**); effect of P7 (at 4 × MIC) on the growth curve of all *S. aureus* strains used in the experiments (strains 18, 189, and 195) followed at 600 nm for a period of 6 h (**b**).

**Table 1 ijms-22-05021-t001:** Main physicochemical features of P7 [20].

Analysis	Feature	Determinations	
FTIR	NH_3_^+^	3500 cm^−1^	
	C-H alkyl	2800–2900 cm^−1^	
	Overtones *	2000–1700 cm^−1^	
	C=ONH	1649 cm^−1^	
	-C=C- *	1575, 1510 cm^−1^	
	*o*-disubstituted ^§^	754 cm^−1^	
VPO ^1^ analysis (MeOH, 45 °C)	Mn	13719	
Volumetric Titration	µequivNH_2_/gP7	305	
DLS ^2^ Analysis	Z-Ave ^3^ (nm)	220 ± 18	
	PDI ^4^	0.809 ± 0.004	
	Z-potential ^5^ (ζ-p)	+49.8 ± 5.8	
Potentiometric Titration	Max dpH/dV ^6^	10.75	4
	HCl 0.1N (mL) ^7^	0.6	1.2
	pH ^8^	6.85	4.80

^1^ Vapor-pressure osmometry; ^2^ dynamic light scattering; ^3^ hydrodynamic diameters of P7 particles; ^4^ polydispersity index; ^5^ measure of the electrical charge of P7 particles suspended in the liquid of acquisition (water); ^6^ max values of first derivative curve of titration curve, indicating the existence of a two-step protonation process; ^7^ volumes of HCl 0.1N needed to protonate P7; ^8^ pH values at which protonations occur; * refers to phenyl rings; ^§^ refers to phenyl rings derived from styrene.

**Table 2 ijms-22-05021-t002:** MIC and MBC values of P7 and of the monomer M7 against bacteria of Gram-positive species, obtained from experiments carried out in triplicate ^1^, expressed as µM and as µg/mL.

	P7 (13719) ^2^		M7 (274) ^2^	
Strains	MIC µM (µg/mL)	MBC µM (µg/mL)	MIC µM (µg/mL)	MBC µM (µg/mL)
***Enterococcus* Genus**				
*E. faecalis* 1 *	2.3 (32)	2.3 (32)	233.6 (64)	467.1 (128)
*E. faecalis* 18 *	2.3 (32)	2.3 (32)	233.6 (64)	467.1 (128)
*E. faecalis* 51 *	2.3 (32)	4.6 (64)	233.6 (64)	467.1 (128)
*E. faecalis* 365 *	2.3 (32)	4.6 (64)	467.1 (128)	467.1 (128)
*E. faecium* 325 *	1.15 (16)	2.3 (32)	116.8 (32)	233.6 (64)
*E. faecium* 341 *	1.15 (16)	2.3 (32)	233.6 (64)	467.1 (128)
*E. faecium* 364 *	0.6 (8)	1.15 (16)	233.6 (64)	467.1 (128)
**Minor Strains**				
*E. casseliflavus* 184 °	1.15 (16)	2.3 (32)	233.6 (64)	467.1 (128)
*E. durans* 103 °	1.15 (16)	2.3 (32)	467.1 (128)	934.3 (256)
*E. gallinarum* 150 *	1.15 (16)	2.3 (32)	467.1 (128)	467.1 (128)
***Staphylococcus* Genus**				
*S. aureus* 18 **	4.6 (64)	4.6 (64)	233.6 (64)	467.1 (128)
*S. aureus* 195 **	4.6 (64)	4.6 (64)	233.6 (64)	467.1 (128)
*S. aureus* 189	4.6 (64)	4.6 (64)	233.6 (64)	467.1 (128)
*S. epidermidis* 22 **	1.15 (16)	2.3 (32)	233.6 (64)	467.1 (128)
*S. epidermidis* 180 ***	1.15 (16)	2.3 (32)	116.8 (32)	233.6 (64)
*S. epidermidis* 181 ***	1.15 (16)	2.3 (32)	233.6 (64)	467.1 (128)
*S. haemolyticus* 193 **	3.15 (16)	3.15 (16)	233.6 (64)	233.6 (64)
*S. hominis* 125 **	0.6 (8)	0.6 (8)	116.8 (32)	116.8 (32)
*S. lugdunensis* 129	1.15 (16)	1.15 (16)	116.8 (32)	233.6 (64)
*S. warneri* 74	1.15 (16)	1.15 (16)	233.6 (64)	233.6 (64)
*S. saprophyticus* 41	1.15 (16)	2.3 (32)	116.8 (32)	233.6 (64)
*S. simulans* 163 **	1.15 (16)	1.15 (16)	467.1 (128)	467.1 (128)
**Sporogenic Isolate**				
*B. subtilis*	1.15 (16)	1.15 (16)	233.6 (64)	467.1 (128)

^1^ the degree of concordance was 3/3 in all the experiments, and standard deviation (±SD) was zero; ^2^ Mn of copolymer P7 and MW of monomer M7; * denotes vancomycin resistance (VRE); ° denotes intrinsic resistance to vancomycin; ** denotes methicillin resistance; *** denotes resistance toward methicillin and linezolid.

**Table 3 ijms-22-05021-t003:** MIC and MBC values of P7 and of the monomer M7 against bacteria of Gram-negative species, obtained from experiments carried out in triplicate ^1^, expressed as µM and as µg/mL.

	P7 (13719) ^2^		M7 (274) ^2^	
Strains	MIC µM (µg/mL)	MBC µM (µg/mL)	MIC µM (µg/mL)	MBC µM (µg/mL)
**Enterobacteriaceae Family**				
*E. coli* 224 S	2.3 (32)	2.3 (32)	467.1 (128)	467.1 (128)
*E. coli* 238 #	4.6 (64)	4.6 (64)	934.3 (256)	934.3 (256)
*E. coli* 246 ^§^	2.3 (32)	2.3 (32)	467.1 (128)	467.1 (128)
*Y. enterocolitica* 342	9.3 (128)	18.6 (256)	467.1 (128)	934.3 (256)
*S. marcescens* 228	>18.6 (>256)	nt ^3^	1868.6 (512)	1868.6 (512)
*P. mirabilis* 254	18.6 (256)	18.6 (256)	1868.6 (512)	1868.6 (512)
*M. morganii* 372	18.6 (256)	>18.6 (>256)	1868.6 (512)	1868.6 (512)
*K. oxytoca* 252	9.3 (128)	9.3 (128)	467.1 (128)	467.1 (128)
*K. pneumoniae* 231 #	4.6 (64)	4.6 (64)	467.1 (128)	467.1 (128)
*K. pneumoniae* 232 #	9.3 (128)	18.6 (256)	467.1 (128)	467.1 (128)
*K. pneumoniae* 233 #	4.6 (64)	4.6 (64)	467.1 (128)	467.1 (128)
*K. pneumoniae* 260 #	9.3 (128)	9.3 (128)	467.1 (128)	467.1 (128)
*K. pneumoniae* 366 #	4.6 (64)	4.6 (64)	467.1 (128)	467.1 (128)
*K. pneumoniae* 367 #	9.3 (128)	9.3 (128)	467.1 (128)	467.1 (128)
*K. pneumoniae* 369	9.3 (128)	18.6 (256)	467.1 (128)	467.1 (128)
*K. pneumoniae* 377 S	4.6 (64)	4.6 (64)	467.1 (128)	467.1 (128)
*Salmonella gr. B* 227	4.6 (64)	4.6 (64)	467.1 (128)	467.1 (128)
*P. stuartii* 374	9.3 (128)	9.3 (128)	1868.6 (512)	1868.6 (512)
**Non-Fermenting Species**				
*A. baumannii* 257	2.3 (32)	4.6 (64)	467.1 (128)	934.3 (256)
*A. baumannii* 279	2.3 (32)	2.3 (32)	467.1 (128)	467.1 (128)
*A. baumannii* 236	9.3 (128)	9.3 (128)	233.6 (64)	467.1 (128)
*A. baumannii* 245	9.3 (128)	4.6 (64)	467.1 (128)	934.3 (256)
*A. baumannii* 383	4.6 (64)	4.6 (64)	467.1 (128)	467.1 (128)
*A. pittii* 272	2.3 (32)	2.3 (32)	233.6 (64)	467.1 (128)
*P. aeruginosa* 229	4.6 (64)	4.6 (64)	>1868.6 (>512)	nt ^3^
*P. aeruginosa* 247	4.6 (64)	9.3 (128)	>1868.6 (>512)	nt ^3^
*P. aeruginosa* 256	4.6 (64)	4.6 (64)	>1868.6 (>512)	nt ^3^
*P. aeruginosa* 259	9.3 (128)	18.6 (256)	>1868.6 (>512)	nt ^3^
*P. aeruginosa* 268	4.6 (64)	4.6 (64)	>1868.6 (>512)	nt ^3^
*P. aeruginosa* 269	2.3 (32)	2.3 (32)	>1868.6 (>512)	nt ^3^
*P. fluorescens* 278	2.3 (32)	2.3 (32)	934.3 (256)	934.3 (256)
*P. putida* 262	4.6 (64)	4.6 (64)	934.3 (256)	934.3 (256)
*S. maltophylia* 255	9.3 (128)	18.6 (256)	467.1 (128)	934.3 (256)
*S. maltophylia* 280	4.6 (64)	9.3 (128)	467.1 (128)	934.3 (256)
*S. maltophylia* 2	2.3 (32)	2.3 (32)	467.1 (128)	467.1 (128)
*S. maltophylia* 3	4.6 (64)	9.3 (128)	467.1 (128)	467.1 (128)
*S. maltophylia* 4	2.3 (32)	4.6 (64)	233.6 (64)	233.6 (64)
*S. maltophylia* 5	9.3 (128)	18.6 (256)	467.1 (128)	467.1 (128)

^1^ the degree of concordance was 3/3 in all the experiments, and standard deviation (±SD) was zero; ^2^ Mn of copolymer P7 and MW of monomer M7; ^3^ means not tested; S denotes susceptibility to all antibiotics; # denotes carbapenemase (KPC)-producing. ^§^ denotes O157:H7; *P. aeruginosa*, *S. maltophylia* and *A. baumannii* are all MDR bacteria.

**Table 4 ijms-22-05021-t004:** MIC values of P7 against the most relevant MDR isolates tested in this study compared to the MIC values of antibiotics commonly used against the same species. MIC values were obtained from experiments carried out in triplicate^1^ and are expressed as µM concentrations.

	P7 (13719) ^2^	Commercial Antibiotics
Strains	MIC (µM)	MIC (µM)
***Enterococcus* Genus**		
*E. faecalis **	2.3	22.1–88.3 ^3^
*E. faecium **	1.15	88.3–176.6 ^3^
***Staphylococcus* Genus**		
*S. aureus ***	4.6	637.7–1275 ^4^
*S. epidermidis ****	1.15	637.7 ^4^
**Sporogenic Isolate**		
*B. subtilis*	1.15	212.4 ^5^
**Enterobacteriaceae Family**		
*E. coli* #	4.6	16.8 ^6^ 96.6 ^7^ 13.3–26.5 ^5^
*K. pneumoniae* #	4.6–9.3	4.4–16.8 ^6^ 96.6–193.2 ^7^ 6.7–13.3 ^5^
*Salmonella gr. B*	4.6	235.5 ^8^
*M. morganii*	18.6	212.4 ^5^
**Non-Fermenting Species**		
*A. baumannii*	2.3–9.3	96.6–193.2 ^7^
*P. aeruginosa*	2.3–9.3	19.6–156.5 ^9^
*S. maltophylia*	2.3–9.3	58.8–117.7 ^8^

^1^ The degree of concordance was 3/3 in all the experiments, and standard deviation (±SD) was zero; ^2^ Mn of copolymer P7; ^3^ vancomycin; ^4^ oxacillin; ^5^ amoxy-clavulanate; ^6^ ertapenem; ^7^ ciprofloxacin; ^8^ trimethoprim-sulfamethoxazole; ^9^ piperacillin tazobactam; * denotes vancomycin resistance (VRE); ** denotes methicillin resistance; *** denotes resistance toward methicillin and linezolid; # denotes carbapenemase (KPC)-producing; *P. aeruginosa*, *S. maltophylia*, and *A. baumannii* are all MDR bacteria.

## Data Availability

All data concerning this study are contained in the present manuscript or in previous articles whose references have been provided.

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
