# Peer review of "Broad-Spectrum Bactericidal Activity of a Synthetic Random Copolymer Based on 2-Methoxy-6-(4-Vinylbenzyloxy)-Benzylammonium Hydrochloride"

_ijms, 2021, doi:10.3390/ijms22095021_

Round 1

Reviewer 1 Report

In the presented work, the Authors characterized the antimicrobial properties of water-soluble non-quaternary copolymeric ammonium salt (P7), whose synthesis had been described in their previous work. The antibacterial activity of the compound was determined toward clinical strains of Gram-positive and Gram-negative bacteria, including drug-resistant microorganisms. Due to the fact that the Authors showed that P7 had high antibacterial activity, a question arises whether they determined  its cytotoxicity to normal mammalian cells and haemolytic activity? The lack of toxicity to eukaryotic cells is a crucial condition that potential drugs must fulfill, therefore it seems necessary to conduct this type of studies

In addition, I would like to ask the Authors to address the following issues:

  • have the MIC and MBC values been determined for commonly used antibiotics? if not, please complete this issue in the discussion based on the literature data
  • Have any studies been performed to demonstrate the mechanism of P7 action?

Author Response

In the presented work, the Authors characterized the antimicrobial properties of water-soluble non-quaternary copolymeric ammonium salt (P7), whose synthesis had been described in their previous work. The antibacterial activity of the compound was determined toward clinical strains of Gram-positive and Gram-negative bacteria, including drug-resistant microorganisms. Due to the fact that the Authors showed that P7 had high antibacterial activity, a question arises whether they determined  its cytotoxicity to normal mammalian cells and haemolytic activity? The lack of toxicity to eukaryotic cells is a crucial condition that potential drugs must fulfill, therefore it seems necessary to conduct this type of studies.

Firstly, we thank the Reviewer for the positive general comments on our study and we hope to further satisfy him/her with our following responses to his/her requests.

Regarding the evaluation of the cytotoxicity and haemolytic toxicity of P7 on mammalian cells, we agree with the Reviewer that such experiments are mandatory, before formulating it in administrable forms, subjecting it to further characterizations and suggesting it for biomedical applications.

Such copious series of obligatory steps requires an appropriate research work (already in progress), which was not within the scope of the present manuscript, that was specifically dedicated to the description and characterization of the antibacterial activity possessed by P7.

For the moment, the evidence reported below (as already in the main text) suggests that P7 is not cytotoxic to mammalian cells.

  • in the design of the synthesis of P7, some suggestions were followed, already reported in the literature, aimed at favoring a good antibacterial activity, and reducing haemolytic toxicity and cytotoxicity towards eukaryotic cells (lines 66-95).
  • as previously reported (Ref. 21 and 25), cationic materials such as P7, with both antibacterial and antitumor activity, exert a disruptive action on bacterial membranes and on those of malignant cells (like each other), being instead harmless on healthy eukaryotes cells, due to the different characteristics of their membranes.

Please, see in the main text at lines 93-101, 119-124 and 529-530 (Ref. 21 and 25 have been added in these lines and are highlighted in light blue).

However, to further underline the probable lack of cytotoxicity of P7 towards eukaryotic cells, due to its action on the bacterial outer envelope different from that of mammalian cells, a new sentence has been added (lines 174-175).

In addition, I would like to ask the Authors to address the following issues:

have the MIC and MBC values been determined for commonly used antibiotics? if not, please complete this issue in the discussion based on the literature data

As requested by the Reviewer, in the revised version of our manuscript, a new Table (Table 4) has been inserted, in which, for each multidrug-resistant bacteria species tested in this study, we reported the single value or the range of MIC values of P7 against the isolates tested. Then, for comparison purposes, we reported MIC ranges of commonly used commercial antibiotics on the same strains of that species, obtained by their antibiograms. Sentences of introduction (lines 329-333) and of comment (lines 343-348) to Table 4 have been also inserted.

Have any studies been performed to demonstrate the mechanism of P7 action?

We thank the Reviewer for his comment, which gives us the opportunity to further highlight the main and widely recognized mechanism of action of cationic materials, toxic to both bacterial and cancerous cells, due to the presence of specific similar constituents in their membranes which are missing in the membrane of normal eukaryotic cells (Please, see Ref. 21 and 25).

As already highlighted in the main text, and as reported in numerous studies cited in the present work (Ref. 11, 20, 22 are only examples of the cited works), the antibacterial effects of cationic peptides, polymers, copolymers, and dendrimers is mainly due to their ability to electrostatically interact with the negative components of bacterial surface, thus causing direct destabilization and enhancing permeabilization through the formation of pores.  After the absorption on the bacterial surface, the cationic agents could intercalate with the membranes and progressively alter their normal structure, cause irreparable damage, and thus determine the loss of enzymes, coenzymes, ions, and cytoplasmic contents that eventually trigger bacterial death.

However, as reported at the end of the introduction (lines 135-137), and in our previous works on antibacterial cationic agents (both dendrimers and polymers, see Ref. 13,23,24), the biocidal properties and the mechanism of action with which P7 acts on bacterial cells have been investigated on opportunely selected strains of both Gram-positive and Gram-negative species by performing time-killing and turbidimetric experiments.

In this regard, a dissertation about the rapid and not-lytic mechanism of action of P7, which agreed with results previously observed, was already reported in the original version of our manuscript in Sections 2.3.3 and 2.3.4.

Reviewer 2 Report

This M.S. sounds interesting.  I have a minor concerns shown below:

  1. How these compounds can inhibit the MDR bacteria  ?  The mechanism can be shown in discussion section.

Author Response

This M.S. sounds interesting.  I have a minor concerns shown below:

How these compounds can inhibit the MDR bacteria?  The mechanism can be shown in discussion section.

The answer to the question of the Reviewer was already present in the original version of the manuscript at lines 101-108.

“Moreover, it was generally accepted, that the very low propensity to induce resistance of these cationic macromolecules derives from their non-specific mechanism of action. In virtue of this, the cationic agents do not need necessarily to enter the cell and target enzymatic processes, that can mutate developing resistance, but they are capable to kill the target rapidly, simply on contact [11,13,22-24]. Consequently, widespread, and rapid resistance development towards membrane active and cell lysing antimicrobial macromolecules is unlikely compared to conventional antimicrobials.”

Anyway, to satisfy the Reviewer, a further explanation concerning the capability of cationic agents to overcoming the resistance of bacteria to traditional antibiotics was added in the Discussion Section. Please, see lines 220-227.